

# Phylogeography of the widespread Caribbean spiny orb weaver *Gasteracantha cancriformis*

Lisa Chamberland[1], Fabian C. Salgado-Roa[2], Alma Basco[3], Amanda Crastz-Flores[4], Greta J. Binford[5] and Ingi Agnarsson[1,6]

[1] Department of Biology, University of Vermont, Burlington, VT, USA
[2] Biology Program, Faculty of Natural Sciences and Mathematics, Universidad del Rosario, Bogota, Colombia
[3] University of Puerto Rico at Rio Piedras, San Juan, Puerto Rico
[4] Universidad Metropolitana (now Ana G. Mendez University), San Juan, Puerto Rico
[5] Department of Biology, Lewis & Clark College, Portland, OR, USA
[6] Department of Entomology, National Museum of Natural History, Smithsonian Institution, Washington, DC, USA

Corresponding author
Lisa Chamberland,
lchambe1@uvm.edu

## ABSTRACT

**Background:** Modern molecular analyses are often inconsistent with pre-cladistic taxonomic hypotheses, frequently indicating higher richness than morphological taxonomy estimates. Among Caribbean spiders, widespread species are relatively few compared to the prevalence of single island endemics. The taxonomic hypothesis *Gasteracantha cancriformis* circumscribes a species with profuse variation in size, color and body form. Distributed throughout the Neotropics, *G. cancriformis* is the only morphological species of *Gasteracantha* in the New World in this globally distributed genus.

**Methods:** We inferred phylogenetic relationships across Neotropical populations of *Gasteracantha* using three target genes. Within the Caribbean, we estimated genetic diversity, population structure, and gene flow among island populations.

**Results:** Our findings revealed a single widespread species of *Gasteracantha* throughout the Caribbean, *G. cancriformis*, while suggesting two recently divergent mainland populations that may represent separate species, diverging linages, or geographically isolated demes. The concatenated and *COI* (Cytochrome c oxidase subunit 1) phylogeny supported a Caribbean clade nested within the New World. Genetic variability was high between island populations for our *COI* dataset; however, gene flow was also high, especially between large, adjacent islands. We found structured genetic and morphological variation within *G. cancriformis* island populations; however, this variation does not reflect genealogical relationships. Rather, isolation by distance and local morphological adaptation may explain the observed variation.

## INTRODUCTION

Tropical island archipelagos are some of the most biodiverse and species-rich ecosystems on the planet (*Mittermeier et al., 2011*). As spatially discrete microcosms, islands are

exemplary models for studying evolutionary patterns and processes (*Hedges, 2001*; *Ricklefs & Bermingham, 2008*; *Hedges, 2001*). While local habitat heterogeneity may generate diverse ecological niches providing opportunities for adaptive radiations (*Gillespie & Roderick, 2002*; *Gillespie, 2004*), barriers between (e.g., oceans) and within (e.g., mountain ranges, rivers, valleys) islands likely operate more rapidly in initial diversification of lineages newly colonizing oceanic archipelagos. The Caribbean islands are a biodiversity hotspot, rich in endemic species (*Mittermeier & Goettsch Mittermeier, 2005*; *Ricklefs & Bermingham, 2008*). Heterogeneous local environments and diverse and time-deep geological histories (*Gillespie & Roderick, 2014*) have generated a kaleidoscope of communities resulting from historical evolutionary and current ecological selection (*Antonelli & Sanmartín, 2011*; *Smith et al., 2014*). The Caribbean Sea is characterized by volcanic activity and mid-ocean ridges along the Great Caribbean arc (*Pindell & Barrett, 1990*; *Pindell et al., 2006*). Darwinian (oceanic) islands (*Gillespie & Roderick, 2002*) are formed de novo often along subduction zones and geologic hotspots; these islands were never connected to the mainland and are surrounded by deep oceanic barriers. Oceanic islands can have both volcanic and sedimentary (e.g., Lesser Antilles, Bermuda) and non-volcanic (e.g., limestone islands of the Bahamas) origins. Species compositions on these islands are characterized by long-distance dispersals (LDD) (*De Queiroz, 2005*; *Cowie & Holland, 2006*; *Gillespie et al., 2012*). Conversely, continental islands once shared ancient subaerial connections to the mainland during periods of low sea levels and have subsequently flooded. The Greater Antilles are hypothesized to have shared an ancient (33–35 My) subaerial connection to South America via the Greater Antilles Aves Ridge (GAARlandia) (*Iturralde-Vinent & MacPhee, 1999*; *Iturralde-Vinent, 2006*), which would have opened passageways for flora and fauna to readily disperse until subsequently diversifying following vicariance events. There is an historical and ongoing debate in the field of biogeography regarding the relative importance of LDD versus vicariance in diversification and distributions of species (*Ali, 2012*; *Agnarsson, Ali & Barrington, 2019*). Contemporary biogeographic studies have revealed that both LDD and vicariance are important for the distributions of many lineages within the Caribbean (*Hedges, 1996*; *Cowie & Holland, 2006*; *Holland & Cowie, 2007*; *Chamberland et al., 2018*; *Čandek et al., 2020*; *Tong, Binford & Agnarsson, 2019*; *Crews & Esposito, 2020*).

Islands have been used extensively as tools for studying dispersal patterns because oceans act as tough filters against taxa with low vagility (*Cowie & Holland, 2006*). A lineage's biogeography is often dependent on a combination of factors including dispersal ability, breadth of habitat suitability, and competition. Higher dispersal propensity typically results in more gene flow and thus lower genetic structure between populations. Alternatively, taxa with poor dispersal capacity often display higher genetic structure with evolutionary histories that more closely mirror geologic histories. Many arachnids are excellent models for studying biogeographic and evolutionary questions. Spiders, in particular, have evolved diverse web architecture and hunting strategies that presumably allowed them to occupy an impressive range of ecological niches (*Blackledge et al., 2009*). They also have a wide range of dispersal abilities—many spiderlings, for example, can disperse over incredible distances by releasing strands of silk into the air that

are then carried off by wind (*Bell et al., 2005*), a process known as 'ballooning'. The majority of Caribbean arachnids that have undergone biogeographic analyses are short-range endemics (*Cosgrove et al., 2016*; *Esposito et al., 2015*; *McHugh et al., 2014*; *Dziki et al., 2015*; *Agnarsson et al., 2018*; *Chamberland et al., 2018*; *Tong, Binford & Agnarsson, 2019*; *Čandek et al., 2020*) with a few widespread species. Wider ranging species are restricted to lineages with high vagility (*Esposito et al., 2015*; *Crews & Gillespie, 2010*; *Cosgrove et al., 2016*; *Agnarsson et al., 2016*) and a handful of species associated with humans (*Crews & Gillespie, 2010*). A relatively small portion of the Caribbean biota represents 'widespread species' shared among islands and with neighboring continental landmasses (*Losos, 1996*; *Ricklefs & Bermingham, 2008*; *Claramunt et al., 2012*; *Dziki et al., 2015*; *Agnarsson et al., 2016*; *Deler-Hernández, 2017*; *Čandek et al., 2020*).

*Gasteracantha*, the spiny-backed orb weaver, is a widespread spider genus (70 species described globally). These spiders build remarkably conspicuous orb webs in open areas (*Levi, 1978*) often affixing these large webs to shrubs, trees, and/or buildings (*Edmunds & Edmunds, 1986*). Their webs are decorated with silk structures, stabilimenta, that alert large animals of their presence thus preventing accidental collisions (*Jaffé et al., 2006*; *Eberhard, 2007*). *Gasteracantha* have colorful abdomens with hard, sclerotized spines. Body coloration among spiders can serve as visual lures for mating (*Li et al., 2008*, *Lim, Land & Li, 2007*) and prey capture (*Hauber, 2002*; *Tso et al., 2006*; *Tso, Huang & Liao, 2007*; *Fan, Yang & Tso, 2009*; *Blamires et al., 2011*; *Rao et al., 2015*; *White & Kemp, 2016*), as well as camouflage (*Foelix, 1982*; *Blackledge, 1998*; *Oxford & Gillespie, 1998*), crypsis or predator avoidance (*Foelix, 1982*). Brightly colored abdomens are common among orb-weavers and have been widely studied. Still, the ecological underpinnings of inter- and intraspecific morphological diversity are quite varied (*Oxford & Gillespie, 1998*) and remain largely unsolved within this genus (*Ximenes & Gawryszewski, 2018*). The spines, which come in pairs of two or three, have been postulated to play a role in predatory defense in the similarly spiny genus *Micrathena* (*Peckham, 1889*; *Edmunds & Edmunds, 1986*; *Cloudsley-Thompson, 1995*; *Gonzaga, 2007*); however, this hypothesis has never been empirically tested.

Profuse morphological variation and broad distributions within *Gasteracantha* have led taxonomists to debate the number of species, particularly within the Caribbean where molecular data has been absent. Since *Linnaeus (1758)* initially described *G. cancriformis*, eight species have been named in the New World (*World Spider Catalog, 2020*; *Taczanowski, 1879*; *Mello-Leitão, 1917*; *Thorell, 1859*; *Butler, 1873*; *Koch, 1844*; *Guérin-Méneville, 1825*; *Strand, 1916*, *Linnaeus, 1767*). *Linnaeus (1767)* and *Wunderlich (1986)* both recognized a possible second four-spined species of *Gasteracantha* in the Americas-*G. tetracantha*. *Levi (1996*, *2002)* synonymized these into a single species, *G. cancriformis*, currently the only recognized species of *Gasteracantha* in the New World (*World Spider Catalog, 2020*).

Here we tested the taxonomic hypothesis and phylogeography of *G. cancriformis*. In particular, our goals were to: (1) use molecular data to reconstruct a novel phylogeny; (2) examine the genetic diversity, population structure, and geographic distribution within

the Caribbean; (3) assess correspondences between genetics, morphology, and geology. To accomplish these goals, we present a novel molecular phylogeny based on the most extensive sampling and first molecular dataset within the Caribbean.

## MATERIALS AND METHOD

### Sampling

*Gasteracantha* were collected (2011–2016) from Cuba, Hispaniola, Jamaica, Mona, Puerto Rico, the Lesser Antilles, Turks and Caicos (TCI), Mexico, Costa Rica and the South Eastern United States (SEUS) using standard aerial searching and vegetation beating methods (*Coddington et al., 1991*). The specimens were fixed in 95% ethanol in the field and stored in −20 °C freezers in the lab. We obtained sequence data from GenBank for our outgroups and for additional South American *Gasteracantha* (data from *Salgado-Roa et al. (2018)*). Our outgroup species included near relatives selected based on recent phylogenetic analyses of Araneidae and Theridiidae spider families (*Bond et al., 2014*; *Dimitrov et al., 2016*; *Garrison et al., 2016*). Taxon sample information and GPS localities are included in Table S1. All specimens were collected under appropriate collection permits and approved guidelines. USDI National Park Service, EVER-2013-SCI-0028; Costa Rica, SINAC, pasaporte científico no. 05933, resolución no. 019-2013-SINAC; Cuba, Departamento de Recursos Naturales, PE 2012/05, 2012003 and 2012001; Dominican Republic, Ministerio de Medio Ambiente y Recur-sos Naturales, no. 0577; Mexico, SEMARNAT scientific collector permit FAUT-0175 issued to Dr. Oscar Federico Francke Ballve, Oficio no. SGPA/DGVS/10102/13; Colombia, Authoridad Nacional de Licencias Ambientales, 18.497.666 issued to Alexander Gómez Mejía; Saba, The Executive Council of the Public Entity Saba, no. 112/2013; Martinique, Ministère de L'Écologie, du Développement Durable, et de L'Énergie; Nevis, Nevis Historical & Conservation Society, no F001; Barbados, Ministry of Environment and Drainage, no 8434/56/1 Vol. II.

### DNA extraction, amplification, sequencing and alignment

We extracted DNA from 148 individuals with the QIAGEN DNeasy extraction kit. We amplified two mitochondrial loci (*COI*: Cytochrome c oxidase subunit I), (*16S*: 16SrRNA), and one nuclear loci (*ITS2*: internal transcribed spacer 2), that have demonstrated successful amplification and informative variation in spiders at low taxonomic levels (*Agnarsson, Maddison & Avilés, 2007*; *Agnarsson, 2010*; *Kuntner & Agnarsson, 2011*; *McHugh et al., 2014*). PCR conditions of the three markers are described in Table S2. DNA sequences were assembled using Phred and Phrap (*Green, 2009*; *Green & Ewing, 2002*) via the Chromaseq module 1.2 in Mesquite 3.61 (*Maddison & Maddison, 2019*) with default parameters. Outgroup sequence data were taken from Genbank (Table S1). All gene fragments were aligned in MAFFT 7 (http://mafft.cbrc.jp/alignment/server/) under default settings. The final alignments were then aligned by eye, edited and maintained in Mesquite.

## Phylogenetics and divergence time estimations

We selected the appropriate substitution model and partitioning schemes using PartitionFinder v1.1.1 (*Lanfear et al., 2012*) using the 'greedy' algorithm and 'mrbayes' model according to the AIC criterion (*Posada & Buckley, 2004*). We used Bayesian inference (BI) to test the phylogenetic relationships and estimate divergence times within Neotropical *Gasteracantha*. We generated individual gene trees for the three loci and for a concatenated dataset remotely on the CIPRES (*Miller, Pfeiffer & Schwartz, 2010*) portal using MrBayes 3.2.2 (*Huelsenbeck et al., 2001*; *Ronquist & Huelsenbeck, 2003*). We used a concatenated phylogeny because it has been shown to perform as well as species tree methods (*Tonini et al., 2015*). Four BI, Markov chain Monte Carlo (MCMC) were run with two sets of four chains for 100 million generations, sampling the Markov chain every 10,000 generations. All tree files were examined in Tracer v1.6 (*Rambaut et al., 2014*) to verify proper mixing of chains and that MCMC had reached stationarity (effective sample size, ESS > 200), and to determine adequate burn-in. The burn-in was set for the first 5,000 trees. We computed posterior probabilities (PP) from a majority rule consensus tree of the post-burn-in trees locally in MrBayes.

Node ages were estimated using BEAST 1.8.0 (*Drummond & Rambaut, 2007*) under a relaxed clock model (*Drummond et al., 2006*, *2012*). We configured input files locally using BEAUti (*Altekar et al., 2004*) and then ran the BEAST analysis on the CIPRES online portal. We pruned sequences that had greater than 65% missing characters as well as redundant sequences with one individual per haplotype to avoid coalescence and zero-length branches. The most recent common ancestor (MRCA) of Araneoidea (Theridiidae + Araneidae) was calibrated using a normal distribution with mean of 170 Ma (SD ± 35) and the age of the root as 233 (*Garrison et al., 2016*). The MRCA of Araneidae was set to 70 My (SD ± 28). The *COI* mitochondrial substitution rate parameter (ucld. mean) was set as a normal prior with mean = 0.0112 and SD = 0.001; these substitution rates have been estimated to be similar across spider lineages (*Bidegaray-Batista & Arnedo, 2011*; *Kuntner et al., 2013*; *McHugh et al., 2014*). The analysis was run for 60 million generations with a calibrated birth-death tree prior as it can simulate extinction rates over time and is more appropriate if more than one individual represents terminal taxa (*Drummond et al., 2012*). For individual gene trees and our concatenated dataset, we assessed convergence of the runs and tested for stationarity (ESS > 200) in Tracer. A maximum clade credibility tree was assembled in TreeAnnotator using a burn-in of 5 million generations for all three loci and for the concatenated dataset.

## Genetic diversity, population structure and haplotype reconstructio

We assessed population structure and estimated genetic differentiation between Caribbean island populations of *G. cancriformis*. Since our main focus was within the Caribbean, we did not include the mainland species/populations for the following analyses. Using each island as an operational geographic unit, we ran an analysis of molecular variance (AMOVA) in Arlequin v. 3.5 (*Excoffier & Lischer, 2010*) to infer hierarchical structure across island populations. We performed two AMOVAs, the first within only island populations of *G. cancriformis* and the second with all of *G. cancriformis* including the

mainland population. Haplotype diversity (Hd) (*Nei, 1987*), pairwise estimates of nucleotide diversity ($\pi$) (*Nei & Tajima, 1981*), and average nucleotide differences (K) were calculated in DnaSP quantify genetic heterogeneity within each island. Using DnaSP, we also estimated relative ($F_{ST}$) and absolute ($d_{XY}$) differentiation between island populations. We calculated *Rousset's (1997)* distance measure ($F_{ST}$)/(1− $F_{ST}$) (100,000 permutations) in Arlequin to test partitioning of genetic variation by islands in *G. cancriformis*. Number of migrants (Nm) was calculated in Arlequin to estimate gene flow between populations. Haplotype networks were assessed using median-joining methods in PopART 1.7 (http://popart.otago.ac.nz) (*Bandelt, Forster & Röhl, 1999*; *Leigh & Bryant, 2015*). We implemented Bayesian Analysis of Population Structure (BAPS) (*Cheng et al., 2013*), a hierarchical genetic clustering algorithm, to assess the nested population structure within *G. cancriformis* using the R-package *RhierBAPS* (*Tonkin-Hill et al., 2018*). We performed two separate runs testing 1–20 populations for both our mtDNA and nuclear datasets.

## Species boundaries

DNA barcoding and genetic distance were used to test for species boundaries within Neotropical *Gasteracantha*. Uncorrected p values among and between potential species groups were calculated in Mega 7 (*Kumar, Stecher & Tamura, 2016*). We tested species boundaries using Automatic Barcode Gap Discovery (ABGD) method (*Puillandre et al., 2012*) rather than generalized mixed Yule-coalescent (GMYC) or Poisson tree processes (PTP) because the latter are not specifically designed to find recently diverged species and are often sensitive to high gene flow (*Luo et al., 2018*). The ABGD method was used through the online portal (http://www.abi.snv.jussieu.fr/public/abgd/) to identify shifts from low intraspecific distances to higher interspecific in the *COI* sequences. We set P (prior intraspecific divergence) from 0.01 to 0.1; steps set to 10; X (minimum relative gap width) set to 1.5; Nb bins (for distance distribution) set to 20; we selected the Kimura (K80) model and set TS/TV to 2.0.

## Geographic, genetic and morphological distances

We tested the relationships between geographic, genetic and morphological (spine number and abdomen color) distances. We ran a Mantel test for 10,000 permutations using the R-package *ecodist* (*Goslee & Urban, 2007*) to assess significance between genetic relatedness and geographic distance. To assess geographic and genetic patterns of polymorphisms, adult female specimens were photographed using Visionary Digital BK lab system. We used 80 adult females from the Caribbean islands for our morphological analyses. We coded a total of six color morphs—three that have been previously described within *G. cancriformis* by *Gawryszewski (2007)* (white, yellow, black and white), an all-black morph previously reported by *Salgado-Roa et al. (2018)*, and two morphs that had not been previously reported (white and red stripes, black and yellow stripes) (Fig. 1). We ran a chi-squared Monte Carlo analysis to test the association between the coloration and spine number of individuals and their genetic variation and geographic locality. Under the null hypothesis we would expect morphology

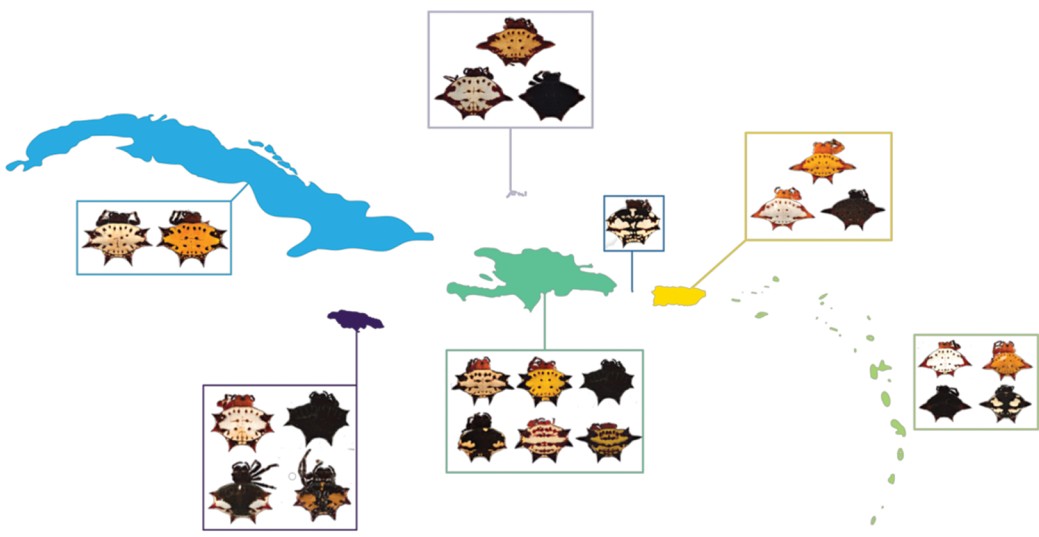

**Figure 1** ***G. cancriformis* phenotypes collected from Cuba, Hispaniola, Puerto Rico, Jamaica, Mona, TCI and the Lesser Antilles.** Four-spined morphs were exclusive to Puerto Rico, the Lesser Antilles and TCI.                                                            

(abdominal color and spine number) to be independent of island. We ran Multiple Correspondence Analysis (MCA) using the R-package *FactoMineR* (*Lê, Josse & Husson, 2008*) to analyze the statistical variance among our three categorical variables (island, *COI* haplotype, and spine number and abdominal color.

# RESULTS

## Phylogenetics and divergence time estimations

The final alignment lengths for our DNA matrices were: 529 – *COI*, 546 – *16S* and 506 – *ITS2*, for a concatenated matrix of 1581 base pairs. The models used in the MrBayes and BEAST analyses for each of the loci were HKY + I + G for *COI*, GTR + G for *16S*, and GTR + I + G for *ITS2*. Phylogenetic inferences for both the single-gene (*COI*) and pruned (42 terminals) BEAST concatenated datasets indicated three clades (PP > 0.75) within the New World—two mainland clades and a predominantly island clade that included some individuals from SEUS (Fig. 2; Fig. S1). The full, 209 specimen concatenated phylogeny was inconsistent with our pruned concatenated phylogeny; the two divergent populations, Caribbean and South America (west of the Andes), were nested a single New World clade. (Fig. S2). We inferred phylogenetic relationships with reference to the pruned, dated topology, because of the implicit biases in in our full dataset including sampling biases, large amounts of missing data for 16S and ITS2 (particularly in Central America), and redundant sequences.

## Genetic diversity, population structure and haplotype reconstruction

There is high genetic diversity, but low genetic divergence among island populations of *G. cancriformis*. The BAPS inference revealed three genetic clusters within *G. cancriformis*: a western cluster containing SEUS, Cuba, Jamaica, TCI, a central cluster containing Hispaniola, Mona and Puerto Rico, and an eastern cluster containing Hispaniola, Puerto
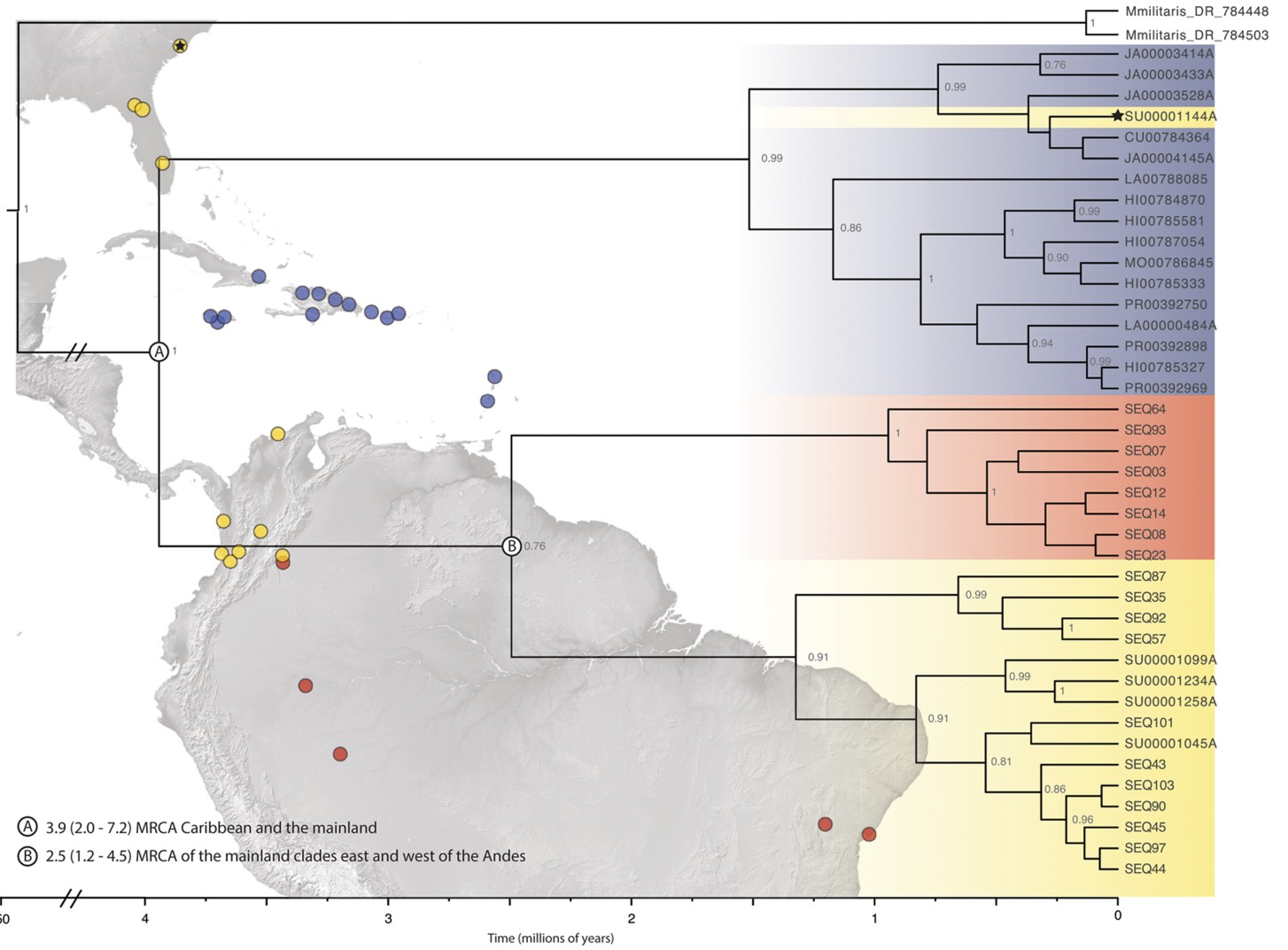

**Figure 2 Beast divergence times estimations of concatenated phylogeny.** Dataset was pruned to exclude redundant taxa and individuals with >65% missing data. Nodes are labeled with BI posterior probability values; any nodes with PP < 0.75 are not labeled. The colors indicate broad geographic location (blue = Caribbean, yellow = North, Central, and South America-west of the Andes, red = South America-east of the Andes) of individuals. There is one specimen within the Caribbean clade from SEUS (indicated by the black star). Major lineage divergence times and 95% highest posterior density of the (A) Caribbean and SEUS clade and (B) mainland clades. (Image credit: https://photojournal.jpl.nasa.gov/catalog/PIA03377; https://photojournal.jpl.nasa.gov/catalog/PIA03388).

Rico and the Lesser Antilles (Table S1). Genetic variability was better explained by differences among islands (49.21%), rather than due to changes among populations within islands (16.75%; Table 1). Pairwise genetic distances and gene flow ranged from −0.00297 to 0.78772 to 0.13475 and infinity between Jamaica and TCI and between Cuba and Mona respectively (Table S3; Fig. 3). Pairwise $F_{ST}$ values were higher between island populations than within populations (Fig. 4C; Table S3). Lowest pairwise genetic distances were between Jamaica and TCI ($F_{ST}$ = −0.00297) with gene flow reaching infinity. The highest pairwise genetic distances were between Mona and Cuba ($F_{ST}$ = 0.78772) and lowest migration rates (Nm = 0.13475) (Table S3). The nucleotide differences ($K_{XY}$) and average number of nucleotide substitutions per site ($d_{XY}$) between

**Table 1 Analysis of molecular variance between and among *G. cancriformis* island populations.**

| Source of variation | Percent variation |
|---|---|
| *COI* | |
| Among islands | 49.21 |
| Among populations within islands | 16.75 |
| Within populations | 34.04 |
| *16S* | |
| Among islands | −0.39 |
| Among populations within islands | 0.75 |
| Within populations | 99.64 |

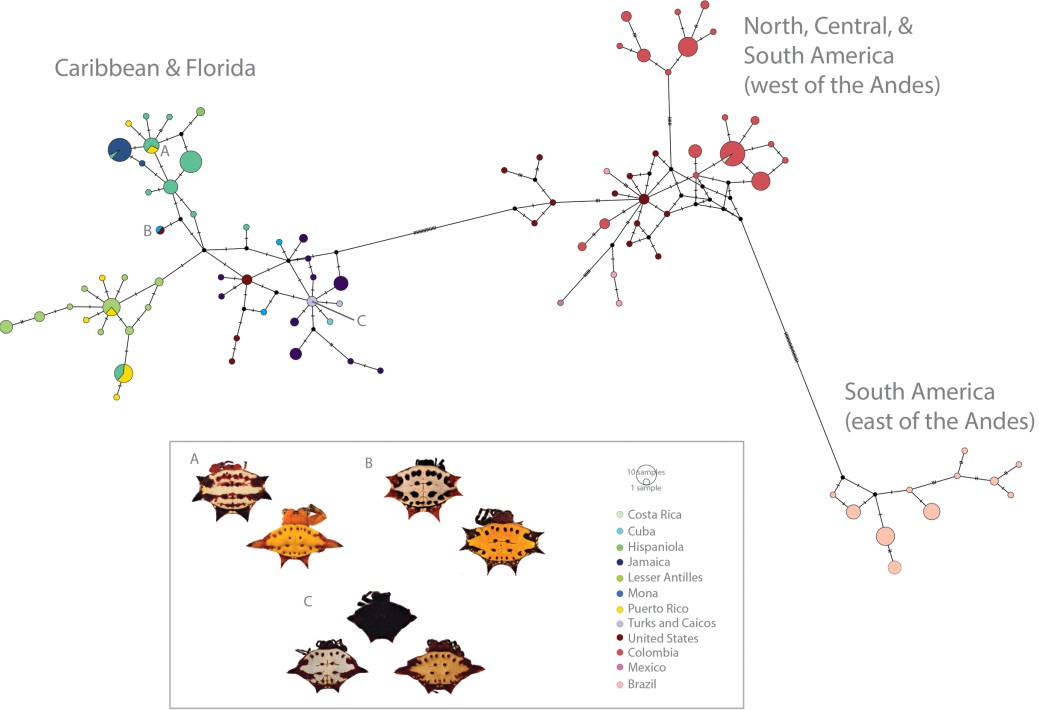

**Figure 3 Haplotype network (*COI*) of *Gasteracantha* collected in the Caribbean and North, Central and South America.** Pie charts are colored by geographic location and are proportional to the number of individuals sharing the haplotype. (A) Polymorphism variation is high even within single islands. (B) color is not specific to island, spines are highly specific to island regardless of phylogenetic placement. (C) Even small islands (e.g., TCI) have multiple colors morphs.

island populations ranged from 2.1 to 0.00397 (Jamaica and TCI) to 7.6875 and 0.01453 (Jamaica and Mona) respectively (Table S4). Within the *COI* dataset, we found 42 unique haplotypes and high haplotype and nuclear diversity (Hd = 0.9473, $N$ = 0.00973 respectively) (Table 2). The average nucleotide differences ($K$) was 5.14855 among island populations. The *16S* dataset had fewer haplotypes ($H$ = 8) and less haplotype diversity

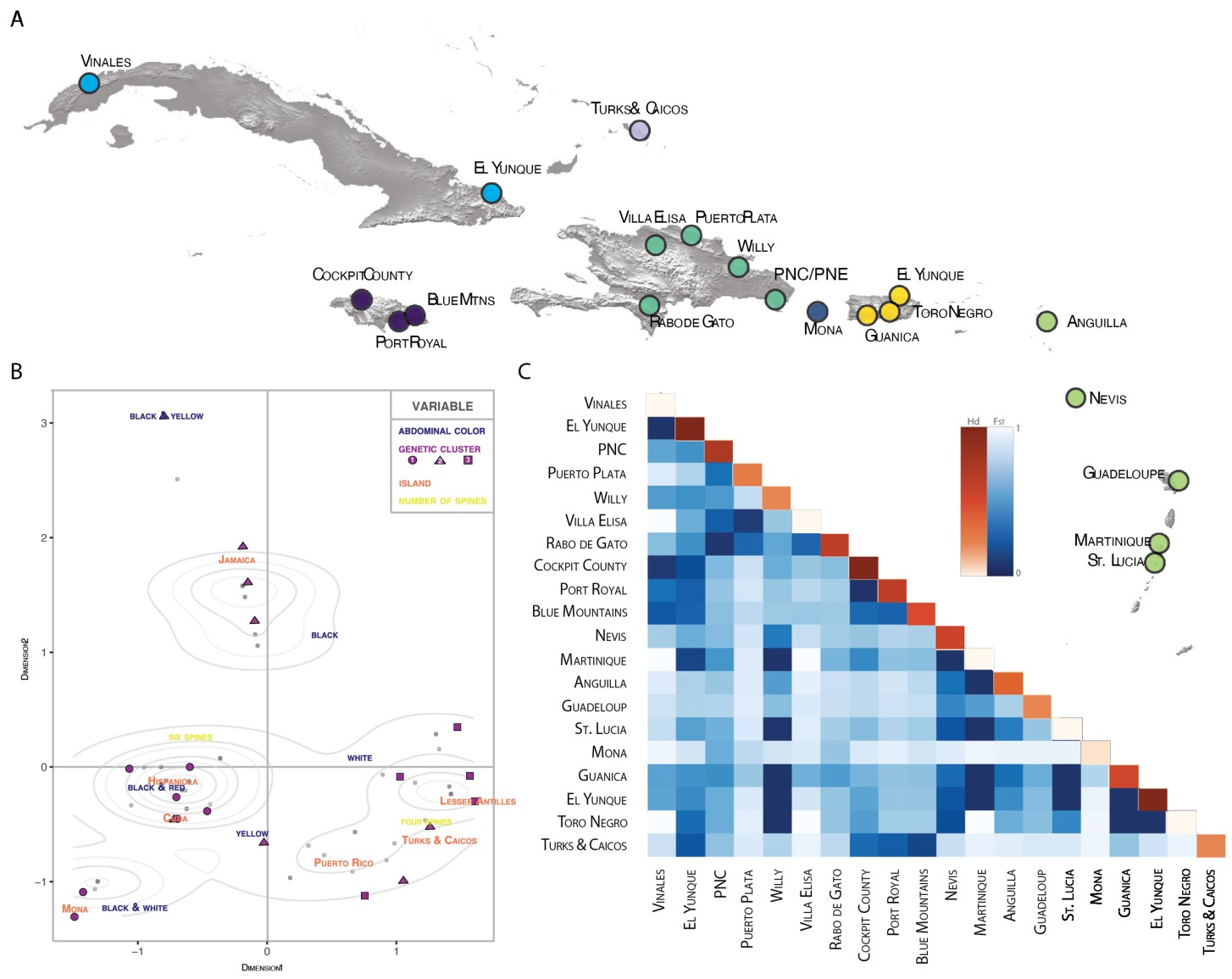

**Figure 4** **Population pairwise genetic diversity of *G. cancriformis* among 20 island populations.** (A) map of major collection localities (B) MCA plot of categorical variables geology (island), genetics (*COI* haplotype), and morphology (spine number and abdomen color). Grey dots represent observations, with darker shaded dots indicating higher concentration of individuals. Density curves surround highly concentrated zones. The first dimension explains 8.1% of the variance in our data and dimension two explains 6.9% of the variance. (C) heatmap of $F_{ST}$ values (blue hues) and within population haplotype diversity across the diagonal (orange hues) of the *COI* Caribbean dataset.

(Hd = 0.406, Nm = 0.00198). Haplotype and nucleotide diversity were relatively lower for islands with smaller areas (e.g., Mona and TCI) (Table 2).

ABGD analyses of *COI* split *Gasteracantha* into three geographic, putative independently evolving groups, Caribbean islands, west of the Andes, and east of the Andes (Fig. 2). Uncorrected *COI* distance p scores between the Caribbean clade and the clades east and west of the Andes were 4.1% and 6.7% respectively and 5.2% between the east and west of the Andes clades. The genetic distances between these three populations are greater than the average maximum intraspecific divergence of 0.96%

**Table 2 *COI* summary statistics for populations of *G. cancriformis* sampled in the Caribbean islands.**

| Island | *N* | *H* | *S* | Hd ± SD | π (pi) | Theta per site from *S*. Theta *W* | Tajimas *D* (D) |
|---|---|---|---|---|---|---|---|
| Cuba | 4 | 4 | 8 | 1 ± 0.177 | 0.00819 | 0.00825 | −0.06867 |
| Hispaniola | 37 | 11 | 16 | 0.815 ± 0.048 | 0.00513 | 0.00725 | −0.9541 |
| Jamaica | 20 | 10 | 14 | 0.879 ± 0.052 | 0.00596 | 0.00746 | −0.73613 |
| Lesser Antilles | 24 | 10 | 9 | 0.870 ± 0.045 | 0.00444 | 0.00456 | −0.08468 |
| Mona | 16 | 2 | 1 | 0.125 ± 0.106 | 0.00024 | 0.00057 | −1.16221 |
| Puerto Rico | 14 | 7 | 11 | 0.813 ± 0.094 | 0.00654 | 0.00654 | 0.00292 |
| Turks & Caicos | 4 | 2 | 1 | 0.500 ± 0.0265 | 0.00095 | 0.00103 | −0.61237 |
| Total | 119 | 42 | 39 | 0.94730 | 0.00973 | $F_{ST}$ = 0.57588 | |
| | | | | | | $K$ = 5.14855 | |

**Note:**
   *N*, number of individuals; *H*, number of haplotypes; *S*, number of segregating sites; Hd, haplotype diversity ± SD; pi, nucleotide diversity ± SD; *D*, Tajima's D.

across another Araneidae genus, *Cyclosa* (*Blagoev et al., 2009*). Uncorrected genetic distances in *COI* data between and among taxon groups. Uncorrected p values calculated among and between groups were compared to previously determined araneae distances- 2.15% mean intraspecific divergence and 6.77% mean divergence between nearest interspecific neighboring taxa groups (*Blagoev et al., 2009*; *Kuntner & Agnarsson, 2011*). A recent study on *Gasteracantha* from Malaysia and closely related sister linages recorded uncorrected p distances from 0.63% to 3.75% within species and 7.95% to 14.60% among; however, these distances were based on a four-loci dataset (*Tan et al., 2019*).

### The relationship between geology, genetics and morphology

We found correspondences between geology, genetics, and morphology (Fig. 4B). First, we found a weak, but significant correlation between geographic and genetic distances. The Mantel test of Caribbean *Gasteracantha* indicated a significant correlation between genetic relatedness and geographic distance (Mantel *r* = 0.5306853, *p*-value = 0.0001). Furthermore, there was a substantial amount of intermixed and shared haplotypes among islands (Fig. 3). Second, we found a significant relationship between both spine number (chi-squared = 101, df = 9, *p*-value = 2.2E−16) and abdominal color (chi-squared = 115.09, df = 30, *p*-value = 6.705E−12) and island. The Lesser Antilles, TCI and Puerto Rico clustered tightly around four spines (Fig. 4B). Both yellow and white color morphs were found on every island excluding Mona, which only had the black and white morph. All of the individuals from SEUS were white with red spines, including the divergent Floridian population. The specimens from Costa Rica were yellow with black spines. Lastly, we did not find a strong correspondence between genetic variation and morphology. Many *COI* haplotypes shared multiple polymorphisms (Figs. 3A–3C). Spine number was also more closely explained by island rather than genetic relatedness. There were instances of shared haplotypes with different spines (Fig. 3); however, each island has either four or six-spined individuals, and there were no islands that shared individuals with both spine numbers (Fig. 1; Fig. S2).
## DISCUSSION

### Overview

Here, we present the first molecular and morphological analyses of *Gasteracantha* in the Caribbean. Using a novel molecular dataset, we find partial support for Levi's taxonomic hypothesis of a single New World species in the genus *Gasteracantha, G. cancriformis* with recently diverged (<5 My) populations in the Caribbean and North and South America (Fig. 2). Bayesian analyses of the full, 209-specimen dataset supported a Caribbean and South American clade (east of the Andes) nested within a paraphyletic lineage (west of the Andes) (Fig. S2). This paraphyly was inconsistent with our BEAST dated phylogenetic analysis, which resolved these three geographic, divergent lineages as monophyletic (Fig. 2). Low genetic divergences and shared haplotypes among islands are consistent with ongoing gene flow and/or recent colonization events and imply high dispersal propensity among this lineage. We also tested whether spine number, abdominal color, or geography could predict the phylogenetic placement of *Gasteracantha* within the Caribbean. Spine number strongly corresponds with island whereas color is less specific to islands, with many islands sharing color morphs (Fig. 1).

### Partial support for Levi's single widespread New World species hypothesis and divergent populations

We found support for a single genetic group of *Gasteracantha* in the Caribbean and evidence for additional divergences in the New World. While the *COI* and concatenated BI supported a divergent mainland and Caribbean lineage, there was discordance among tree topologies between our pruned and full dataset. Thus, we did not find strong support for new species, and we failed to reject Levi's single New World species hypothesis. Our full concatenated phylogeny indicates two divergent populations nested within a mainland population from North and South America west of the Andes (Figs. S2 and S3). More specifically, the nesting of the entire New World lineage within Mexico could suggest origins in Mexico and subsequent diversification in the Caribbean and North and South America, Mexico only included *COI* data and was not included in our pruned phylogeny. Further sampling throughout Central America, would undoubtably be important for uncovering the biogeographic history of the genus in the Neotropics. Furthermore, posterior probabilities were extremely low (Fig. S3), which is likely due to missing sequence data (Table S1) and/or identical sequences. Large amounts of missing data in conjunction with incomplete lineage sorting (ILS) can generate inconsistent and conflicting phylogenies (*Xi, Liu & Davis, 2016*). There were three diagnosable clades within the dated pruned (42 terminal) concatenated and *COI* phylogeny, one Caribbean clade, and two mainland clades (Fig. 2). Our concatenated and *COI* BEAST analysis indicated three recently divergent (<5 Ma) clades-one primarily Caribbean clade (with a few individuals from SEUS) (PP = 1; Fig. 2A), one from North, Central, and South America (west of the Andes), and one clade containing individuals east of the Andes (Orinoco, Amazon and southeastern Brazil) (PP = 76; Fig. 2B). The Andes are a large mountain range that are important drivers of diversification in South American lineages

(*Brumfield & Capparella, 1996*; *Antonelli et al., 2009*; *Hoorn et al., 2010*), including in ferns (*Testo, Sessa & Barrington, 2019*), freshwater fish (*Lundberg et al., 1998*), and have restricted gene flow within *Gasteracantha* (*Salgado-Roa et al., 2018*). Similarly, the Caribbean Sea is likely also playing a role in the divergence of the Caribbean *Gasteracantha*. While we do find evidence of migration and gene flow between Caribbean and North American populations, the oceans in between are likely limiting gene flow, but not so much as to generate new species.

There was discordance among our three individual gene trees. While the *16S* phylogeny showed similar topology to *COI* supporting two mainland clades and a Caribbean and SEUS clade, *ITS2* did not recover these three clades (Fig. S1). Discordance among gene trees may be an artifact of the recent divergences between island and mainland populations or a product of male-biased dispersal (*Pusey, 1987*; *Knight et al., 1999*; *Aars & Ims, 2000*; *Doums, Cabrera & Peeters, 2008*), which would lead to different evolutionary histories between males and females. However, low information content of ITS2—lack of evidence—in this study is a more likely explanation for this discordance. Sexual size dimorphism can be extreme among spiders (*Kuntner & Elgar, 2014*). Females within *G. cancriformis* have more prominent spines and are larger (5–9 mm long, 10–13 mm wide) than the tiny (2–3 mm long) males (*Muma, 1971*; *World Spider Catalog, 2020*). Furthermore, females remain sedentary in the web, while males will disperse and search for potential mates. More likely, however, these discordant gene trees are a consequence of differing substitution rates (*Maddison, 1997*; *Degnan & Rosenberg, 2009*). While we find *COI* structure, the discordant gene trees and evidence of gene flow suggests this is more consistent with recent colonization or founder effects rather than speciation.

Species delimitation analyses revealed genetically divergent populations, uncorrected *COI* distance p scores ranged from 4.1% to 6.7% between island and mainland populations; however, this was for a single gene and discordance among our phylogenetic analyses do not provide strong enough evidence for new species. Population level Next Generation Sequencing will clarify relationships and would help to recover a more robust species tree. Furthermore, future intra-specific studies on this lineage may consider more closely related outgroups based on recent Araneidae phylogenies (*Scharff et al. 2020*; *Kallal et al. 2018*). Still, given that we sampled only a portion of the distribution of *G. cancriformis* (N. America and the Caribbean) additional species may be expected.

**Genetic diversity, population structure, and geographic distribution**

*Gasteracantha* characterize a good dispersing lineage of spiders that colonized the Caribbean over water and have continued exchange among islands. Gene flow is high among island populations of *Gasteracantha* ($G_{ST}$ = 0.22; Nm = 0.88); however, it does not entirely obfuscate a geographic signal. The largest proportion of genetic variability (49.21%) is explained between islands for *COI*. While *16S* did have genetic clustering between the two mainland populations and the islands, it did not did not have significant genetic variability (−0.39%) between the island populations (Table 2; Fig. S4) This may be due to ILS or limited molecular Caribbean data for *16S* and *ITS2*, with 29 and 26

individuals respectively. Moderate genetic structure and evidence of gene flow among island populations implies good dispersal propensity among this lineage of spiders. Similar patterns of widespread distribution are found in good dispersers (*Van der Pijl, 1982*) including flying animals (*Weeks & Claramunt, 2014*), freshwater shrimp (*Cook, Page & Hughes, 2012*), saltwater resistant seeds (*Stephens, 1966*), and ballooning arachnids (*Kuntner & Agnarsson, 2011*; *Agnarsson et al., 2016*). Furthermore, since *Gasteracantha* colonized the Caribbean much later than GAARlandia (33–35 My) would have existed, we reject the land bridge hypothesis for this lineage. The intermediate dispersal model (*Agnarsson, Cheng & Kuntner, 2014*) predicts lineages with the intermediate dispersal abilities will have the greatest species richness. Taxa are able to disperse to islands but remain isolated enough for speciation to occur. Excellent dispersers will exhibit high gene flow among populations and consequently, their phylogeny will not be reflected in the geologic history. Gene flow within the Caribbean populations of *Gasteracantha* has likely prevented speciation on these islands.

## The relationship between geology, genetics and morphology

Individuals that were further apart geographically had greater genetic distances between them. Results from our Mantel test detected a significant correlation between geographic and *COI* genetic distances (Mantel $r = 0.5306853$, $P = 0.0001$). Notably, one major pitfall associated with Mantel tests is that they are often subject to spatial autocorrelation and erroneously low *p*-values (*Guillot & Rousset, 2013*). Still, this relationship between geography and genetics was also reflected in our BAPS cluster analysis (Table S1). Individuals that were geographically close also cluster within the *COI* phylogeny, indicating a correlation between geography and genetics, albeit relatively weak.

We found evidence that spine number was an important predictor of island locality and was highly specific to island. For instance, six-spined and four-spined *Gasteracantha* did not occur on the same island and neither are monophyletic (Fig. 1; Fig. S2). Excluding Mona, six-spined varieties were only found on the mainland and on large islands (e.g., Cuba, Hispaniola), and four-spined varieties were typically found on the smaller islands (e.g., Lesser Antilles, TCI) (Fig. 1). Results from chi-squared analyses support a strong correspondence of spine number with geography (individual islands), but spine number only weakly corresponds with genetic similarity. Even among shared haplotypes between Puerto Rico and Hispaniola the morphology (number of abdominal spines) corresponded with geography—indicating strong selection following island colonization (Fig. 3A). Strong regional selection can drive two genetically different lineages towards a single phenotype (e.g., Müllerian mimicry in *Heliconius* butterflies) (*Hines et al., 2011*; *Supple et al., 2013*). Within *Gasteracantha*, we hypothesize that there are strong selective pressures, namely predators, on each island driving spine number and coloration, as suggested by recent ecological studies (*Ximenes & Gawryszewski, 2018*). Birds (*Rypstra, 1984*; *Wise, 1995*; *Foelix, 1996*) and wasps (*Wise, 1995*; *Foelix, 1996*; *Camillo & Brescovit, 2000*; *Camillo, 2002*) are common predators of spiders. Given that *Gasteracantha* remain exposed in the center of the web during the daytime and birds are primarily visual predators, it is likely spines play an important role in defense. Even with their hard,

sclerotized exterior, *Gasteracantha* are still often consumed by wasps (*Camillo, 2002*, *Gawryszewski & Motta, 2012*; I. Agnarsson, 2020, pers. obs.). Future ecological studies could test whether there is a correlation between endemic species of known bird predators on an island and the number of spines.

In contrast to patterns in spine number, dramatic color polymorphisms were widespread throughout the Caribbean and were generally (excluding Mona) weakly geographically structured. Moreover, we did not find a correspondence between genetics and color morphs. For instance, identical *COI* haplotypes sometimes shared multiple color morphs (Fig. 3C), thus implying strong ecological selection following island colonization. Morphological changes, including adaptive radiations, on islands can occur rapidly following isolation events (*Millien, 2006*). Selective pressures, founder and priority related effects, and local conditions, including temperature, solar radiation, and predators can drive differential environmental adaptations between populations (*Mathys & Lockwood, 2011*). Environmental heterogeneity, such as that exemplified in the Caribbean islands (e.g., variation in light spectrums, island habitat diversity and environmental filtering) can also independently drive disruptive selection for discrete polymorphisms (*Endler, 1993*, *Oxford & Gillespie, 1998*, *Oxford & Gillespie, 2001*). The two genetically divergent populations from SEUS are morphologically identical—all specimens are white with six red spines, a morph previously described in South America by *Gawryszewski (2007)*. Remarkably, one such morph from SEUS, which is nested within the Caribbean clade (Fig. 2), also shares an identical *COI* haplotype with a yellow morph from Cuba (Fig. 3B). This suggests that possible ecological plasticity rather than phylogenetics is driving and or maintaining these highly variant polymorphisms. It is possible that habitat diversity and heterogeneity is responsible for generating and maintaining the dramatic polymorphisms within island populations of *G. cancriformis*. Sampling in North and Central America was limited in this study and is important for more confidently addressing this question. Furthermore, since color polymorphisms are often represented in a small number of major loci (*Ford, 1940*; *Chouteau et al., 2017*; *Gautier et al., 2018*), uncovering the genetic underpinnings of these color polymorphism in *Gasteracantha* will be fundamental in testing the underlying selective pressures of these colors.

Color polymorphisms are paradoxical in nature; typically, genetic drift and natural selection remove variation from populations (*Ford, 1964*; *Hartl & Clark, 1997*; *Futuyma, 2005*). From the predator's perspective, there is a high cognitive demand for detecting cryptic species (*Bond & Kamil, 2002*); thus, predators will invest in searching for one morph (*Poulton, 1890*; *Tinbergen, 1960*; *Bond, 2007*); selection will drive towards one state (*Mallet & Joron, 1999*; *Lehtonen & Kokko, 2012*). Apostatic selection (*Paulson, 1973*), a negative frequency-dependent selection in which rarer morphs are selected upon less than expected until the predator switches to the common form (*Poulton, 1890*; *Tinbergen, 1960*; *Clarke, 1962*; *Allen & Clarke, 1968*; *Allen, 1988*; *Oxford & Gillespie, 1998*; *Bond, 2007*) may be driving these color patterns in the Caribbean, has been postulated to explain the many different colors in tropical insects (*Rand, 1967*). *T. grallator*, the

Hawaiian happy-face spider, are a classic example of discrete color polymorphisms among spiders. *Oxford & Gillespie (2001)* discovered multiple drivers of these polymorphisms on islands and found that many of them have evolved de novo on the islands. Gene flow and genetic drift can also play a fundamental role in maintaining polymorphisms (*Fisher, 1930*; *Ford, 1975*) as sub-optimal morphs can persist in populations with immigrating new individuals providing opportunities for ongoing gene flow (*Gray & McKinnon, 2007*). The proposed hypotheses however are not mutually exclusive; in most instances, it is the interplay between natural selection and genetic drift that maintains genetic diversity and polymorphisms within a population (*Slatkin, 1973*; *O'Hara, 2005*; *Saccheri et al., 2008*; *Iserbyt et al., 2010*). Future studies can tease apart these potential drivers of this remarkable diversity within *G. cancriformis*.

## CONCLUSIONS

We found that *G. cancriformis* includes a distinctive genetic group that is largely from the Caribbean islands, which suggest that the sea is a geographic barrier that promotes genetic differentiation between islands and mainland. However, little evidence of genetic divergence between Caribbean islands, and lack of Caribbean monophyly are indicative of gene flow between islands and continents, suggesting that this species' high vagility facilitates dispersal across geographic barriers. Though we found the presence of unique phenotypes in some islands, the loci that we used did not elucidated an association between patterns of genetic diversity and phenotypic diversity. *Gasteracantha cancriformis* is an ideal model system for future studies to explore ecological, evolutionary, and behavioral questions. The striking polymorphic phenotypes within this species include characters that are closely associated with geography (spine number) and characters that are not (color), leaving much to learn about ecological and behavioral factors that influence the evolutionary maintenance of this color polymorphism. NGS and/or RADseq data would provide higher resolution for testing population level relationships as well as patterns of gene flow and migration between islands.

## ACKNOWLEDGEMENTS

We thank all the members of the CarBio team. We are especially grateful to Anne McHugh, Carol Pfeiffer, and Laura Caicedo for their help collecting specimens. We would also like to thank Morgan Southgate and Federico Lopez-Osorio for their help with the data analyses. We would also like to thank David Barrington for providing feedback on an early version of the manuscript.

### Funding

This work was supported by the National Science Foundation (DEB-1314749 and DEB-1050253) to Ingi Agnarsson and Greta Binford. Additional funds came from the Smithsonian Laboratories of Analytical Biology, a 2013 SI Barcode Network to Jonathan Coddington and Ingi Agnarsson, from Slovenian Research Agency (ARRS) grants to

Matjaz Kuntner. The funders had no role in study design, data collection and analysis, decision to publish, or preparation of the manuscript.

### Grant Disclosures
The following grant information was disclosed by the authors:
National Science Foundation: DEB-1314749 and DEB-1050253.
Smithsonian Laboratories of Analytical Biology.
Slovenian Research Agency (ARRS).

### Competing Interests
The authors declare that they have no competing interests.

### Author Contributions
- Lisa Chamberland conceived and designed the experiments, performed the experiments, analyzed the data, prepared figures and/or tables, authored or reviewed drafts of the paper, and approved the final draft.
- Fabian C. Salgado-Roa conceived and designed the experiments, performed the experiments, authored or reviewed drafts of the paper, and approved the final draft.
- Alma Basco performed the experiments, authored or reviewed drafts of the paper, and approved the final draft.
- Amanda Crastz-Flores performed the experiments, authored or reviewed drafts of the paper, and approved the final draft.
- Greta J. Binford conceived and designed the experiments, authored or reviewed drafts of the paper, and approved the final draft.
- Ingi Agnarsson conceived and designed the experiments, authored or reviewed drafts of the paper, and approved the final draft.

### Field Study Permissions
The following information was supplied relating to field study approvals (i.e., approving body and any reference numbers):

All material was collected under appropriate collection permits and approved guidelines. USA, USDI National Park Service, EVER-2013-SCI-0028; Costa Rica, SINAC, pasaporte científico no. 05933, resolución no. 019-2013-SINAC; Cuba, Departamento de Recursos Naturales, PE 2012/05, 2012003 and 2012001; Dominican Republic, Ministerio de Medio Ambiente y Recursos Naturales, no. 0577; Mexico, SEMARNAT scientific collector permit FAUT-0175 issued to Dr. Oscar Federico Francke Ballve, Oficio no. SGPA/DGVS/10102/13; Colombia, Authoridad Nacional de Licencias Ambientales, 18.497.666 issued to Alexander Gómez Mejía; Saba, The Executive Council of the Public Entity Saba, no. 112/2013; Martinique, Ministère de L'Écologie, du Développement Durable, et de L'Énergie; Nevis, Nevis Historical & Conservation Society, no. F001; Barbados, Ministry of Environment and Drainage, no. 8434/56/1 Vol. II.

## Data Availability

The code is available at GitHub: https://github.com/lchamberland/scripts.

The data is available at GenBank (Supplemental File) and at Dryad: Chamberland, Lisa et al. (2020), Phylogeography of the widespread Caribbean spiny orb weaver *Gasteracantha cancriformis*, v2, Dryad, Dataset, DOI 10.5061/dryad.sf7m0cg2s.

## Supplemental Information

Supplemental information for this article can be found online at http://dx.doi.org/10.7717/peerj.8976#supplemental-information.

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
