# Peer review of "Phylogeography of the widespread Caribbean spiny orb weaver Gasteracantha cancriformis"

_PeerJ, doi:10.7717/peerj.8976_

## Round 0.1 · original submission · Minor Revisions

Your manuscript has now been seen by two reviewers, both of which recommended publication with some modest number of revisions. Reviewer 1 notes some editorial corrections as well as a critique of outgroup choice. As pointed out by the reviewer it's probably not necessary to rerun analyses but it is worth noting that there may be more suitable choices. This reviewer also points out that the reference section needs work. Please also look carefully over the comments by Reviewer 2, but acknowledge that the reviewer seems to have overlooked the analyses examining population structure and genetic/geographic distance.

When you revise your paper, please document in your cover letter how each of the reviewer's suggestions were addressed in your reworked manuscript. Thanks for submitting a really nice manuscript to PeerJ - I look forward to seeing a revised manuscript in short order.

·

Basic reporting

I found the presentation of the paper clear. The introduction is thorough, the methods are complete, and the results and discussion do not draw any inappropriate inferences from the data. The references require special mention as a weak point (see general comments).

Experimental design

The experiments seem largely well thought out and relevant to the questions being asked. My only issues relate to the selection of outgroups and calibration. I elaborate in the general comments but including a theridiid and some of the earlier diverging araneids makes little sense to me in a study focusing on Gasteracantha cancriformis.

Validity of the findings

The results of the paper are well grounded, and where there are questionable relationships (e.g., low posterior probability for Andean split), the authors discuss the possibilities roundly. I think this paper lays the groundwork well for both an expanded phylogeny of gasteracanthine araneids as well as another data point in a future CarBio synthesis of arachnid island biogeography.

Additional comments

Abstract
Background – why is Gasteracantha cancriformis set off by commas?

Introduction
L30 – 'championing' implies intent, no?
L82 – No mention of stabilimenta?
L83 – bold text needs to be corrected
L94 – 'upwards of eight' – surely we can more specific?

Methods
Your outgroup selection is curious. Argiope, Zygiella, and Micrathena are not particularly closely related to Gasteracantha based on recent molecular phylogenies focusing on the family using Sanger (Scharff et al. 2019) or high throughput methods (Kallal et al. 2018), both of which are unmentioned in lieu of older or less specific studies. The theridiid, separated from araneids by nearly 200 million years and the most distantly related family to Araneidae, is also a strange thing to include for an intraspecific study. Of course, I wouldn't recommend rerunning analyses, but reconsidering the relevant outgroups could be relevant for future similar studies from this working group.

L136 – By 'inferring sequences,' do you mean assembling contigs?
L138 – Was MAFFT used with default settings for all three loci? If so, say so; if not, elaborate – especially given COI may behave differently since it is protein-coding.
L161 – Aranoidea > Araneoidea
L170 – Did you also test for stationarity in BEAST by observing the results in Tracer?
L211 – What are those six color morphs? How are they defined? I know there is a citation but it might help to describe them all the same

Results
L223 – I think 'terminals' instead of 'specimens'

Discussion
L307 – 16S and ITS topologies not discussed much in comparison to COI
L313 – undoubtable > undoubtably
L320 – but Andean split with low support.
L336 – WSC date 2020
L355 – Remove comma after 16S

Conclusions
L451 – sea, rather than ocean (I realize they are tantamount as far as dispersal is concerned)
L455 – No comma required after 'Though'
L457 – Do not begin a sentence with an abbreviated genus name.

References
The references are a mess. I went through A-E with over a dozen issues. Please double check them all.

In ref list, not in main text:
Agnarsson et al 2019
Allen 1988
Altekar et al 2004
Bell et al 2005
Boughman et al 2002
Clarke 1962a
Clarke 1962b
Eberhard 2007
Endler 1979

In text but not in ref list:
Allen & Clarke 1968
Bell 2005
Čandek et al 2019
Clarke 1962
Drummond et al 2006

Furthermore, the following could be useful to discuss or at least mention:

Crews & Esposito (2020) Towards a synthesis of the Caribbean biogeography of terrestrial arthropods. BMC Evol Biol
Seems very relevant and they discuss trends relating to past work from this working group.

Tan et al (2019) Phylogenetic relationships of Actinacantha Simon, Gasteracantha Sundevall, Macracantha Hasselt and Thelacantha Simon spiny orbweavers (Araneae: Araneidae) in Peninsular Malaysia. Raffles Bulletin
It includes taxa that would have been relevant to you at the time and discusses the fraught systematics and biogeography of the subfamily.

Figures
Very nice overall. I think fig 4 has some overlay issues. That is, there is something overlapping the left half of B as well as part of a horizontal line overlapping part of the map, most obviously where it crosses Nevis and a horizontal line above Guadeloupe.

Reviewer 2 ·

Basic reporting

no comment

Experimental design

no comment

Validity of the findings

no comment

Additional comments

The manuscript presents a very good data set of a spider species that occurs in the Americas; the samples comprise the Caribbean islands and North and South America. Methods are well described; data are not yet public but are provided and are very well presented. Literature is well presented as well.

Considering the title, the data set and the the quality of the sample design and of the results, I would suggest the authors to do some reformulations on the hypothesis, with some additional analysis that might improve the interest in the article. I am convinced that the data are suitable to address hypothesis about the colonization and populational dynamics of this spider species in the Americas; my suggestions are mainly about the analysis of the molecular data; the morphological data are very well explored.

The title is “Phylogeography of the widespread Caribbean spiny orb weaver Gasteracantha cancriformis” and, indeed, the sample design is focused on Caribbean islands, including significant representatives from other regions. However, the abstract starts with the discrepancies between molecular and morphological taxonomic estimates and the goals (lines 102-107) sound exploratory; in addition, the discussion (line 287)l begins with “… first phylogenetic and morphological analyses…”.

For my good surprise, along ‘Introduction” a very interesting question is presented concerning the history and colonization of Caribbean region (see lines 47-49). So, having in mind that the genus Gasteracantha has many species but just one in the New World, a preliminary question could be the time of origin of G.cancriformes, or “when it arrived” (vicariance or dispersal, old or recent), and it would need just to include a couple of other Gasteracantha species as outgroups in the dated phylogeny. Anyhow, the diversification of the lineages are so recent (less than 4MY) that is very difficult to talk about speciation (this might lead to a discussion about speciation x incipient diversification, depending on the time of this species branch, and to consider the meaning of the patterns of morphological variation in a recent scenario).

Next, I would like to suggest a populational approach to the data. Populational data are not easy to collect and the authors have such a nice set of samples that deserve to be more explored with phylogeographical methods. The genetic structure is isolation by distance, suggestive of a dispersion process, but the Mantel test can also be applied for each of the three lineages; when there is no IBD, the genetic structure can be tested using a software like BAPS, for example (http://www.helsinki.fi/bsg/software/BAPS/). Estimates of populational (or lineages) expansion/retraction can be explored with bayesian skyline plots; these analyses will complement the descriptions of the branching topology and, together with the haplotypes network, can generate hypotheses about the pathways (and timing) of the colonization of G.cancriformis in the Americas that can be tested with approximate Bayesian computation for model selection (eg. Pelletier & Carstens 2014 - https://doi.org/10.1111/mec.12722).

Minor suggestions:
Methods – please explain why the data was concatenated and how this was done (instead of partitioning the different data sets in a same inference)
Fig 2 – present the time intervals of the nodes (and consider them in discussion).
Table S3 – please indicate confidence intervals for FST

---

## Round 0.2 · accepted · Accept

Thanks for the careful attention to all of the reviewers' comments. It's a great manuscript and I look forward to seeing it published.